# Does a school-based intervention increase girls' sexual and reproductive health attitudes and intentions? Results from a mixed-methods cluster-randomized trial in Burkina Faso

**Laura Hinson**[1]*, **Triantafyllos Pliakas**[2,3], **Emily Schaub**[1], **Aya Mahaman Nourou**[4], **Cecelia Angelone**[5], **Mohamad Ibrahim Brooks**[5], **Abdou Arthur Abga**[6], **Zakari Congo**[6], **Bruno Ki**[6], **Reshma Trasi**[7]

1 International Center for Research on Women, Washington, DC, United States of America, 2 Department of Public Health, Environments and Society, Faculty of Public Health and Policy, London School of Hygiene and Tropical Medicine, London, United Kingdom, 3 GSK Vaccines, Wavre, Belgium and Impact Epilysis, Thessaloniki, Greece, 4 REM Africa, Niamey, Niger, 5 Pathfinder International, Washington, DC, United States of America, 6 Pathfinder International, Ouagadougou, Burkina Faso, 7 Trasi Duarte Consulting, Santa Clara, CA, United States of America

* lshinson23@yahoo.com

**Data Availability Statement:** Location/name: Harvard Dataverse DOI or link: https://dataverse.

## Abstract

Adolescent girls in Burkina Faso face unintended pregnancy risk due to a lack of contraceptive use. The (re)solve project was designed to address contraceptive misperceptions and increase girls' perceptions of their pregnancy risk, primarily through a participatory game and a health passport aimed at easing health facility access. The intervention components were implemented for girls in private and public school in grades 4ème and 3ème (grades 9 and 10) in Ouagadougou and Bobo Dioulasso, Burkina Faso. We conducted an impact evaluation using a mixed-methods cluster randomized control trial design to evaluate (re)solve's impact on girls' intentions to use contraception, among other outcomes. Thirty-two schools were randomly allocated intervention or control. The evaluation included quantitative longitudinal surveys at baseline (N = 2,372) and endline (N = 2,072), qualitative in-depth interviews with girls in the intervention group at baseline (N = 41) and endline (N = 48) and with implementation staff (N = 35) and experts (N = 14) at endline. We used generalized estimating equations (GEE) analysis for the main analysis. Girls receiving the intervention had more positive attitudes related to contraception at endline compared to girls at control schools. (re)solve had a positive effect on girls' intention to use contraception (aOR = 1.59, 95% CI 0.97–2.61), though this did not reach statistical significance. The impact was particularly large among girls who had never had sex, girls who attended public schools, and girls in 3ème. Qualitative findings suggest the intervention was well received and positively shifted attitudes and facility-seeking behaviors for many girls, but that myths and misconceptions related to contraceptive use persist in this mostly young, sexually naïve population. That the (re)solve intervention appears to have shifted adolescent girls' attitudes toward using contraception, coupled with positive trends in intention to use contraception, indicates

harvard.edu/dataset.xhtml?persistentId=doi:10.7910/DVN/GM0YP5.

**Funding:** This manuscript was supported by the (re)solve Project funded by the Bill & Melinda Gates Foundation, Investment ID INV-009344. LH, TP, ES, AMN, CA, MIB, AAA, ZC, BK, and RT all received funding from this source. The funders had no role in study design, data collection and analysis, decision to publish, or preparation of the manuscript.

**Competing interests:** The authors have declared that no competing interests exist.

that interventions like (re)solve may 'prime the pump' for behavior change and increasing girls' use of contraception.

**Trial registration number and date:** https://www.isrctn.com/ISRCTN15387847 Registered on June 15th 2021.

## Introduction

Globally, among adolescent girls aged 15 to 19, 52% have an unmet need for contraception [1]. In Burkina Faso adolescent girls experience high rates of unintended pregnancy, at roughly 99 births per 1,000 girls aged 15 to 19 [2]. Contraceptive use among young women in Burkina Faso is extremely low, with 83% of women aged 15 to 24 who have had sex having never used a modern contraceptive method. The median age at first contraceptive use for urban Burkinabé women ages 25–49 is 22, four years after having sex for the first time (median age 18.1) and over a year after getting married (median age 20.9) [3]. Almost a quarter of nonusers of contraceptives aged 15 to 49 who wanted to delay their next birth by two or more years said they were not using contraceptives because they were not married, and 47.9% did not believe themselves to be at risk of pregnancy or in need of contraceptives [4].

Despite decades of sexual and reproductive health (SRH) programming in Sub-Saharan Africa (SSA), there is a dearth of rigorous evidence on the impact of school-based programming. Most of the existing literature has focused on HIV/AIDS programming measuring outcomes such as condom use. Several literature and systematic reviews on school-based interventions for SRH in SSA report positive changes in attitudes but weak and inconsistent findings on behavior change [5–9]. For example, in the most recent review, two studies showed significant decline in unintended pregnancies and four showed improvements in HIV/AIDS-related knowledge, condom use, and attitudes toward testing; however, data on sexual transmitted infection outcomes and sexual debut were incomplete, unreliable or did not show statistically significant results [9]. One study from Ghana found that the addition of school-based and outreach activities to community mobilization and provider training efforts increased service usage for adolescents, including STI, perinatal and antenatal services [10]. There are few if any studies assessing school-based SRH activities in West Africa, and very few in SSA focused explicitly on adolescent pregnancy rather than HIV/AIDS.

The (re)solve project (described in detail below) was designed to address reasons for non-use of modern contraceptives among unmarried, in-school adolescent girls in 4ème and 3ème (grades 9 and 10 respectively) in Burkina Faso. The primary objective of the intervention, designed through (re)solve's multi-phased process, was to demystify and normalize contraception as a valid option of pregnancy prevention for girls. Other objectives of the project included providing accurate information about contraception, menstruation, puberty and sexual activity and encouraging girls to follow up with questions at a health facility, with the ultimate goal of method uptake and pregnancies averted. The (re)solve intervention was designed to be "light touch" and a scalable behavioral change program that could shift SRH attitudes among adolescent girls in a low-resource setting.

Data generated through the (re)solve behavioral diagnosis process (Through a segmentation analysis and behavioral diagnosis process, the team identified behavioral segments and prioritized events, situations or cognitive patterns of thinking (i.e. "behavioral bottlenecks") that block girls from taking actions that would otherwise meet their needs.) suggested that the main reasons explaining girls' nonuse of contraception in Burkina Faso primarily included

that girls do not explicitly think about all the consequences of sex; perceive they are at low risk of getting pregnant; do not think contraception is intended for girls like them; fear that others will find out they are using contraception; and have more appealing options to avoid pregnancy than contraception [11]. Studies from sub-Saharan Africa confirm these factors impede access to and demand for contraception among young people [12, 13]. These reasons were theorized to be driven by features like attitudes, beliefs and norms that interfere with the decision-making process along the pathway from intention formation (e.g. the decision to become pregnant or not, or to use a method) to follow-through (e.g. at a clinic, the decision to get a method). Knowing that actual contraception use would be unlikely achievable during the six months implementation timeframe, the project team focused instead on influencing attitudes related to pregnancy risk and contraception, and intention to use contraception, which is commonly used in the field and is often predictive of future use [14–17].

This paper presents data from the mixed-method evaluation of a novel school-based adolescent SRH intervention in Burkina Faso. Specifically, we assessed whether the (re)solve package of solutions had an impact on girls' intention to use contraception in the near future, among other attitudinal outcomes such feelings and perceptions about modern contraceptives.

## The (re)solve intervention

The (re)solve intervention was informed by several processes that generated data and insights on young women's barriers to contraceptive use and nonuse in Burkina Faso. Through behavioral landscape analysis and behavioral diagnosis, the project team identified behavioral segments (i.e., subgroups that share characteristics like needs or demographic profiles) through segmentation analysis and prioritized behavioral bottlenecks (i.e., aspects that prevent individuals from making decisions or taking actions). The combined insights from segmentation analysis and behavioral diagnosis into individual, social, and structural barriers served as primary inputs into the design of the intervention. The insights and the intervention components both informed the identification of the specific subpopulation of unmarried schoolgirls in 4ème and 3ème among whom the interventions were user tested [11]. The solutions were also tested with health providers and program implementers.

This collective feedback informed the final content, design, and implementation of the intervention, which comprised the following components [11]:

- **School-based, participatory board game *(La Chance)***: The game was played in a classroom by three teams of two schoolgirls each from 4ème and 3ème (grades 9 and 10 respectively), and was facilitated by a trained community health worker. Game play typically lasted about an hour and was held during lunch break or free periods. Through the board game, girls explored a series of real-life relationship scenarios, increased their pregnancy risk perception, and confronted SRH myths and misconceptions. To advance play, students engaged in strategic decision making, answered trivia questions, and responded to discussion prompts. After each game, the facilitator led a discussion on what the girls learned and encouraged the girls to talk to a health care provider.

- **Health Facility Passport**: After the game, the facilitators gave girls passports that listed health facilities where providers were familiar with the (re)solve intervention. Arriving at the facilities with a health passport signaled girls' interest in SRH information, counseling, or services, including contraception, to facility staff. Participating health care professionals had been oriented to the intervention and its rationale and had received training on how to provide youth-friendly services. Health workers could therefore see the passports and recognize the services the girls were interested in and quickly provide them with discreet SRH counseling or services.

- **Posters and Name Tags:** Posters that matched the design of the board game were displayed in participating schools and health facilities advertising puberty-related, non-contraceptive counseling and services for girls (such as those related to menstruation). In schools, these posters encouraged girls to access services, and while waiting at the health facility, the posters normalized their presence and gave them reasons to be there should a family member or anyone else see them or ask about the purpose of their visit. Providers and staff trained in youth-friendly services also wore matching name tags to help them be easily identifiable to girls visiting the facilities.

The program team implemented the (re)solve intervention in 16 randomly selected secondary schools—eight each in Bobo-Dioulasso (Bobo) and Ouagadougou (Ouaga) –in 4ème and 3ème. In each location, they trained 16 community-based facilitators to lead the board games and distribute the passports and identified 18 health facilities (nine each in Bobo and Ouaga) to participate. A total of 3,120 girls in grades 3ème and 4ème played *La Chance* between December 2019 and March 2020. Facilitators distributed 11,908 passports to girls in this timeframe.

## Materials and methods

### Evaluation design and hypotheses

We conducted an impact evaluation using a mixed-methods cluster randomized control trial design to evaluate whether the (re)solve package of solutions changed girls' intentions to use contraception, among other attitudinal outcomes. The trial took place among girls primarily aged 14 to 18 years in 4éme and 3éme grade in 32 public and private secondary schools in Ouagadougou and Bobo-Dioulasso in Burkina Faso. The impact evaluation component consisted of quantitative longitudinal surveys at baseline, at two months (midline, in intervention schools only) and seven months (endline) post intervention initiation. Our primary outcome of interest was intention to use contraception within the next three months. Other outcomes of interest included attitudes related to SRH, such as the belief that contraception causes infertility. Attitudes are important individual markers of underlying belief systems and are often correlated with intention and behavior, including attitudes about family planning with contraceptive use [18, 19]. The data were analyzed according to the intention to treat principle.

Our primary hypothesis was that girls in 4ème and 3ème who were exposed to the board game and given a health passport to facilitate follow-up at health facility would be more likely to report positive attitudes toward and accurate knowledge about sex and contraception and form an intention to use contraception.

We also conducted qualitative in-depth interviews (IDIs) with adolescent girls at baseline and endline in the intervention schools only. At endline, we also held IDIs with implementation staff and conducted key-informant interviews (KIIs) with experts and authorities in sexual and reproduction health who were not part of the (re)solve project. Through IDIs and KIIs, we sought to contextualize our quantitative findings with a deeper understanding of girls' behaviors and attitudes toward sex and contraception, as well as their perceptions of the attitudes held by peers, adult family members and others in the community.

### Selection of (re)solve health facilities, schools, and participants

We used a multi-stage cluster design with schools as clusters. In the first stage, the team purposively selected nine health facilities in Ouaga and 10 in Bobo from a list of eligible facilities. The purposive sampling was used to ensure that we had diversity of health facilities based on facility type and geographical spread, and to make sure that these facilities all offered youth

friendly services. Next, the team randomly selected schools in the catchment areas of these health facilities. To reach a sample size of 2,400 girls, we selected 32 schools total (16 in each city). We randomly assigned the (re)solve intervention to half of the schools, and the other half were control schools, such that there were eight intervention schools in each city. The intervention was randomly assigned to a higher proportion of private schools, as fewer public schools in Bobo met the criteria. Therefore, we used four public schools and twelve private schools each in Ouaga and Bobo. Of these, half were randomly assigned the intervention and half, the control. We used a random number generator in Excel to assign the first set of schools on each list to intervention and those that followed to control.

At intervention schools, we invited all girls that were interested in participating in the (re)solve intervention to be part of the study. Once the full list of interested girls was prepared, the team randomly sorted the list and then invited the first set of girls on the list to participate in the quantitative survey, followed by the qualitative study. At control schools, all girls in relevant grades interested in participating in the research were put on a list and that list was randomly sorted and the girls whose names were at the top of the list were invited to participate.

## Sample size and power analysis

The research team estimated a sample size of 2,497 girls for the baseline and endline longitudinal impact evaluation assuming (a) 50% of the baseline population would have an intention to use contraception (the most conservative estimate, but also the prevalence of intention to use contraception from a previous project survey among a similar group of adolescent girls in Burkina Faso), (b) a minimal detectable change between baseline and endline of 10%, (c) a design effect of 2.0, (d) a 53% non-response rate on the outcome of interest (based on previous project data, assuming girls who are not sexually active would have different intentions to use contraception), (e) 10% attrition of participants between baseline and endline, and (f) a two-arm design with alpha 0.05 and 80% power. We rounded slightly down to 2,400 to ensure a minimum number of 75 girls involved from each school could be reached given the smallest school in (res)solve had an estimated 75 girls in relevant grades.

## Data collection procedures

The research team collected baseline data between November 2019 and January 2020, after a seven-day training in Ouagadougou for the research partners. After a refresher training, the research team collected midline data in person in January and February 2020. The research team collected endline data in July 2020. Though data collection was originally scheduled for March 2020, it was delayed due to COVID-19 and conducted entirely by phone. At baseline all surveys with girls were conducted in private locations in or around their homes or schools. At endline, all data collection activities took place over the phone due to COVID-19. Due to the unusual nature of phone-based interviews at endline, we provided additional training and spent additional time piloting the tools before implementing the final round of data collection.

A team of mixed-gendered consultants from REM Africa who were experienced in social research and evaluation conducted the interviews. Assessments of the first tranche of data did not show any differences in key outcomes and sensitive questions by gender of the interviewer. The team received a multi-day ethical and project training from ICRW prior to each round of data collection. The team carried out a pilot of all tools and questions based on prepared scripts prior to data collection. When interviewed, participants were requested to remain in a private location where they could not be overheard by anyone else, and to alert the interviewer if someone else entered the room. Interviews lasted approximately 60–90 minutes.

Qualitative interviews were audio recorded, and interviewers also took notes during the interviews. All endline interviews were conducted over the phone due to COVID-19.

## Consent and assent procedures

Under the guidance of CEIRSS, we set the age of majority at 20 years. Therefore, all participants aged 20 and older signed a consent form; for girls younger than 20, we obtained parental consent and then girls' assent for participation. During the assent and consent processes, participants and, when relevant, their parents were made aware that the researcher was part of the REM Africa team, working with ICRW, and conducting a study related to (re)solve.

## Ethics statement

The ICRW Institutional Review Board, based in Washington, DC, reviewed and approved all initial and modified versions of this study, as did the *Comite D'Ethique Institutionnel Pour la Recherche en Sciences de la Sante (CEIRSS)* in Burkina Faso. All members of the research team completed certifications in ethical training.

## Variables for quantitative analysis

Our primary outcome (percent of girls with an intention to use contraception in the next three months) was originally measured using a four-point Likert scale (1 = Yes, definitely; 2 = Yes, probably; 3 = No, probably not; and 4 = No, definitely not). We collapsed this to a binary variable (No/Yes) and included the 12 girls at endline who responded that they preferred not to answer in the "No" category. We chose intention to use contraception in the next three months as our main outcome of interest because we predicted that it was achievable to influence in the timeframe available, unlike other relevant behaviors such as contraceptive use.

Other key variables focused on SRH attitudes, which were focused on individual beliefs about what is true or untrue. Specifically, we measured the percent of girls who strongly agree or agreed that (a) contraception causes infertility ("Modern contraception can cause infertility"), (b) contraception is the best option ("If I am having sex and want to avoid pregnancy modern contraception is best option"), (c) they have the confidence to both get and use contraception ("I feel confident in my ability to get a contraceptive method, if I wanted to avoid pregnancy" and "I feel confident in my ability to use a contraceptive method, if I wanted to avoid pregnancy"), and(d) health care workers do not like to give contraceptive advice to unmarried girls ("Health care workers do not like to give advice to young unmarried girls about family planning"). We collapsed these variables originally using a four-point Likert scale (1 = Strongly agree; 2 = Agree; 3 = Disagree; and 4 = Strongly disagree) into a binary variable (0 = Disagree and 1 = Agree).

Other variables of interest included city, school type (private or public), grade (4eme or 3eme), and age. We used principal component analysis to generate a household wealth index using 10 variables that were collected in the baseline survey among all girls. These included access to electricity, a TV, pay for service TV, personal computer, bicycle, a car or van, a bank account, livestock, internet at home and a motorcycle or scooter. For our analyses, we standardized the index (using the mean and standard deviation of the raw values) and collapsed into quintiles, with the first and fifth quintiles reflecting least and most wealthy households, respectively.

## Statistical analysis

For the main analysis, we used data at endline from 2,072 girls. We assessed baseline characteristics by arm as well as a comparison between the girls that were available at endline for the

main analysis and those girls that dropped out of the study. We present data related to exposure to the intervention, including information about girls' experiences at the health facilities. We also show trends between baseline and endline in the community-level geometric means for primary and other variables, as well as the percentage of girls who reported ever going to a health facility for SRH information or services among the intervention-school girls.

For the main analyses, we examined the association between key sociodemographic and attitudinal predictors and the primary outcome within the regression framework described above. We used generalized estimating equations (GEE) analysis to examine the impact of the intervention on our primary outcome of interest. We developed four models, all of them adjusted for baseline values of the outcome. In the first model (unadjusted model) we report the crude estimate of the impact of the intervention on our outcome of interest. For the adjusted analyses, we first fit models adjusted for age (adjusted Model 1). We then additionally adjusted for predictors that were found to have a statistically significant association with our primary outcome (e.g. COVID-19's effect on mobility, adjusted Model 2). In the final model we additionally adjusted for predictors that were selected *a priori* based on theory and previous research (e.g. city, grade, and wealth quintile, adjusted Model 3). We fit all models with exchangeable correlation matrix (to allow for correlations between participants within the same cluster) and robust standard errors, as recommended for cluster randomized trials with more than 15 clusters per arm [20].

In sensitivity analyses, we ran our models stratified by several variables for which we thought the impact might vary by, including reporting ever having had sex, type of school (i.e. private or public), grade and whether girls in the intervention schools visited a health facility. We also examined the impact of the intervention restricted to girls in the intervention schools who reported or did not report seeking more information and/or contraceptive services at a health facility. Finally, we looked at the impact of the intervention for girls who intended to go a facility but did not follow through because of reasons such as COVID-19 restrictions or a lack of time (i.e. the "near misses"). We examined for interaction effects between intervention arm and the variables above, including key variables like attitudes, as we assumed that the impact of the intervention might be different based on responses to attitudes towards contraception use. These models were adjusted for age, city, COVID-19's effect on mobility and wealth quintile. We report results as odds ratios with 95 percent confidence intervals. *A priori* alpha level was set at 0.05; all analyses were two-sided.

## Qualitative component

Selection, recruitment and consent processes for girls participating in IDIs followed the same procedures as those taking part in the quantitative survey. We randomly selected 4éme and 3éme girls from intervention schools who were interested in participating in the research and obtained parental consent where necessary and girls' assent or consent. At endline, we re-interviewed the same girls who had participated at baseline, where possible. Seventeen of the 48 girls who participated at baseline either were not able to be reached at endline or declined to participate a second time and were replaced by 10 new participants.

We purposively identified implementation staff as participants. Implementation staff included game facilitators and health facility staff familiar with (re)solve. KII participants included health-facility managers, school principals, parent association members, Ministry of Education and Health representatives, and local Pathfinder staff. The research and program teams together determined the most suitable participants.

In IDIs, girls were asked about their knowledge of contraceptive methods, their perception of and attitudes toward adolescent pregnancy and sexuality in their communities, their

personal experience with sex and contraceptive use, and, at endline, their experience with the activities of the (re)solve solutions. KII and (re)solve staff participants were asked about their perceptions of girls' sexual and reproductive health as well as their experience with the (re) solve program. The research team collaboratively developed interview guides for all participant types based on best practices for SRH research and expertise related to the research and cultural context.

Following data collection at baseline and endline, the research team transcribed and translated recordings of all interviews. Transcripts were not returned to participants. Both the English translations and the French transcripts were sent to ICRW, where the research team reviewed them for clarity and quality. The verbatim transcripts were analyzed using NVivo 11, coded by a team of three researchers from ICRW and Pathfinder International. The team developed codes based on the objectives of the qualitative research, namely understanding girls' own sexual activity and use of contraception as well as others' perceptions of and attitudes toward these things. Intercoder reliability was conducted on approximately 15 percent of transcripts for each interview type. Two members of the research team read and coded randomly selected transcripts of each type, then compared for percentage agreement and Kappa coefficient until agreement was deemed sufficient–in most cases above 95 percent for the majority of nodes. After all the transcripts were coded, the research team reviewed code reports to identify common themes. The focus of qualitative analysis was on girls' experience with the intervention, as well as lessons learned and actions taken as a result of the interventions. However, we also include observations from baseline interviews with girls as a point of comparison, particularly around attitudes toward sex and contraception. We also draw from interviews with key informants and implementation staff where relevant, particularly where they have commented on the impacts of the program itself related to our quantitative outcomes.

## Results

We first present the quantitative data followed by a quantitative section contextualizing the qualitative findings organized by two key themes: implementation experiences and attitudes toward contraception.

### Quantitative findings

**Fig 1** shows the analytical sample flow by study arm for the quantitative component. We collected data from 2,372 girls at baseline (1,200 in the control and 1,172 in the intervention schools), 1,144 girls at midline in intervention schools only (98 percent retention) and 2,072 girls at endline (1,054 in the control (89 percent retention) and 1,018 in the intervention schools (87 percent retention).

At baseline, our sample was well-balanced on key demographic characteristics and SRH-related behaviors (**Table 1**). Results indicate few differences between control and intervention-school girls on key SRH-related intentions and attitudes with one exception. In addition, there were few differences at endline between those who dropped out and those who stayed in, suggesting that dropout was random. Two statistically significant differences were noted: dropout girls were much more likely to be from Bobo than Ouaga, and from private schools than public schools (**Table A in S1 Text**).

### Exposure to the (re)solve intervention

Of girls participating in the (re)solve research and programming (N = 1,013), 96.2 percent (N = 974) reported ever playing the game, and 96.7 percent (N = 950) received a passport. The

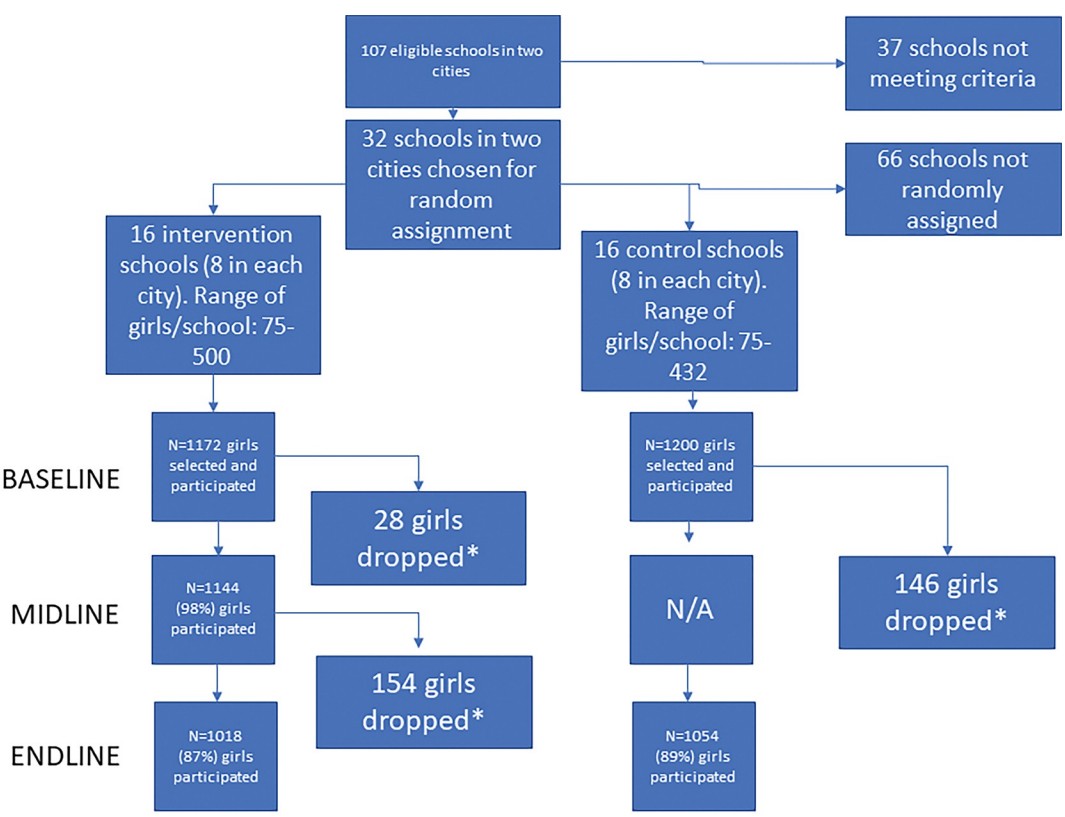

**Fig 1. Flowchart of analytic sample.**

majority received either two (97.2 percent, N = 803) or more than two (14.1%, N = 143) passports to give to other girls, as was intended. The majority of girls (41.9%) reported giving at least one passport to a peer at a different school, followed by an older family member (29.9%). Ninety-one percent (N = 923) saw the posters in school. At endline, 28.7 percent (N = 291) of intervention-school girls had *ever gone* to a health facility for information or services related to puberty or menstruation, and 20.1 percent (N = 204) had *ever done so* for contraceptive information or services.

Of the 204 girls who had ever gone to a health facility for contraceptive information or services, 194 girls (of 1,013, 19.2 percent of total) reported having gone to a health facility *in the last seven months* (during (re)solve implementation) for contraceptive information or services: 14.5 percent in Ouaga (N = 75) and 23.9 percent in Bobo (N = 119) (**Table 2**).

## SRH intentions and attitudes

**Table 3** below show the trends between baseline and endline in the community-level geometric means for the primary outcome as well as other attitudes variables.

We consistently found that girls in the intervention schools shifted their contraceptive intention and attitudes in desired ways, more than control-school girls. For example, there was an absolute shift between baseline and endline of 21.1 in the community-level prevalence in the intervention schools of girls reporting that they had the confidence to get and use contraception, as compared to a shift of only 10.8 among control-school girls.

**Table 1. Participant socio-demographic characteristics and SRH behaviors at baseline.**

| | | Control [N = 1,200] | Intervention [N = 1,172] | Total [N = 2,372] |
|---|---|---|---|---|
| | | N (%) | N (%) | N (%) |
| City | Ouagadougou | 600 (50.0%) | 572 (48.8%) | 1,172 (49.4%) |
| | Bobo-Dioulasso | 600 (50.0%) | 600 (51.2%) | 1,200 (50.6%) |
| School | Private | 900 (75.0%) | 872 (74.4%) | 1,772 (74.7%) |
| | Public | 300 (25.0%) | 300 (25.6%) | 600 (25.3%) |
| Grade | 4ème | 587 (48.9%) | 505 (43.1%) | 1,092 (46.0%) |
| | 3ème | 613 (51.1%) | 667 (56.9%) | 1,280 (54.0%) |
| Age | 14 | 206 (17.2%) | 247 (21.1%) | 453 (19.1%) |
| | 15 | 279 (23.3%) | 281 (24.0%) | 560 (23.6%) |
| | 16 | 269 (22.4%) | 275 (23.5%) | 544 (22.9%) |
| | 17 | 232 (19.3%) | 197 (16.8%) | 429 (18.1%) |
| | 18 | 214 (17.8%) | 172 (14.7%) | 386 (16.3% |
| Wealth | Lowest | 267 (22.3%) | 245 (20.9%) | 512 (21.6%) |
| | Middle-low | 274 (22.8%) | 236 (20.1%) | 510 (21.5%) |
| | Middle | 226 (18.8%) | 229 (19.5%) | 455 (19.2%) |
| | Middle-high | 220 (18.3%) | 231 (19.7%) | 41 (19.0%) |
| | Highest | 213 (17.8%) | 231 (19.7%) | 444 (18.7%) |
| **Behaviors** | | | | |
| % reporting they are currently sexually active | | 96 (8.0%) | 97 (8.3%) | 193 (8.1%) |
| % reporting ever having had sex | | 165 (13.8%) | 169 (14.4%) | 334 (14.1%) |
| % reporting they are currently using contraception | | 44 (3.4%) | 46 (3.9%) | 90 (3.8%) |

Standardized wealth quintiles were created using presence or absence of four household assets variables and validated using principal component analysis. As the allocation of schools between arms is carried out randomly, and girls in each schools were randomly selected, any differences that do occur are assumed to be by chance.

## Main analysis

Our analysis of the association between individual predictors and our primary outcome of interest, intention to use contraception in the next three months (**Table B in S1 Text**), revealed that only a small number of sociodemographic variables were associated with intention, while most of the attitudes toward contraception use were significantly associated with intention to use in the next three months.

The results of our main analyses looking at the impact of (re)solve on intention to use contraception in the next three months are presented in **Table 4**, stratified among various subgroups of girls.

First, the intervention had a positive impact on intention to use contraception among the full sample but did not reach statistical significance in any of the adjusted models, or when stratifying the analysis by sexual activity. Across the whole population, girls who received the (re)solve intervention had higher odds of reporting intention to use contraception in the next three months compared to girls not receiving the intervention (aOR = 1.59, 95% CI 0.97–2.61). We observed similar trends for girls who reported ever having sex (aOR = 1.43, 95% CI 0.79–2.59). Girls in the intervention schools who reported never having sex were almost twice likely to report an intention to use contraception in the next three months compared to girls in the control schools (aOR = 1.80, 95% CI 0.95–3.42). We found no evidence of interaction between arm and having ever been sexually active on having an intention to use contraception in the next three months (p for interaction = 0.685). Please refer to **Fig A and Table C in S1 Text** for results across the different models and stratifications (including coefficients from the adjusted variables).

**Table 2. Experience of girls at (re)solve health facilities.**

| | | Total N (%) |
|---|---|---|
| Saw posters at health facility (N = 194) | | 178 (91.8%) |
| Method received at health facility, as reported by girls (N = 194) | Modern method[a] | 36 (18.6%) |
| | Condoms[b] | 13 (6.7%) |
| | Abstinence | 58 (29.9%) |
| | Other methods or preferred not to respond[c] | 87 (44.8%) |
| Reasons for not visiting health facility for contraception information or services (among the N = 809 girls who did not go) | Not sexually active | 331 (40.9%) |
| | Already using a method | 15 (1.9%) |
| | Not interested | 66 (8.2%) |
| | Intended to but busy with school | 210 (26.0%) |
| | Intended to, but COVID-19 | 89 (11.0%) |
| | Intended to but (other reasons)[d] | 66 (8.2%) |
| | Other, don't know, or prefer not to respond | 32 (4.0%) |

[a]Modern methods include all hormonal methods and emergency contraception.

[b]Condoms include male and female condoms.

[c]Other methods include 67 girls reporting "prefer not to answer."

[d]Other reasons include lack of means, transport, and support.

**Table 3. Trends in SRH intentions and attitudes.**

| | Control | | Intervention | |
|---|---|---|---|---|
| **Variables** | **Baseline** | **Endline** | **Baseline** | **Endline** |
| | **n/N (%*!)** | **n/N (%*)** | **n/N (%*)** | **n/N (%*)** |
| *Intentions* | | | | |
| Intention to use a method of contraception at any time in the next 3 months | 131/1200 | 122/1054 | 133/1172 | 144/1018 |
| | (10.2) | (8.4) | (9.3) | (11.7) |
| *Attitudes* | | | | |
| I feel confident in my ability to use and get a contraceptive method, if I wanted to get pregnant | 989/1200 | 664/1054 | 920/1172 | 759/1018 |
| | (51.7) | (62.5) | (52.2) | (73.3) [+] |
| Modern contraception causes infertility | 880/1200 | 886/1054 | 852/1172 | 785/1018 |
| | (85.3) | (83.8) | (80.9) | (76.4) [+] |
| If I am having sex and want to avoid pregnancy modern contraception is best option | 630/1200 | 806/1054 | 629/1172 | 851/1018 |
| | (73.6) | (75.8) | (72.2) | (82.7) [+] |
| Health care workers do not like to give advice to young unmarried girls about FP | 450/1200 | 381/1054 | 437/1172 | 276/1018 |
| | (37.1) | (33.8) | (36.0) | (25.7) |

*Geometric means across communities.

[+]Indicates a statistically significant difference between control and intervention schools at p<0.05 using the Wilcoxon rank-sum test with 1000 Monte Carlo permutations.

**Table 4. (re)solve's impact on Intention to use contraception among various participants.**

| | n/N (%[a]) | Unadjusted model OR/95% CI | Adjusted model 1 OR/95% CI | Adjusted model 2 OR/95% CI | Adjusted model 3 OR/95% CI |
|---|---|---|---|---|---|
| **All girls (N = 2,072)** | | | | | |
| Control | 122/1054 (8.4%) | 1.00 | 1.00 | 1.00 | 1.00 |
| Intervention | 144/1018 (11.7%) | 1.23 (0.76–1.97) | 1.20 (0.69–2.08) | 1.47 (0.92–2.34) | 1.59 (0.97–2.61) |
| **Ever sexually active girls only (N = 391)** | | | | | |
| Control | 85/211 (38.5%) | 1.00 | 1.00 | 1.00 | 1.00 |
| Intervention | 84/180 (45.1%) | 1.32 (0.75–2.33) | 1.27 (0.71–2.25) | 1.38 (0.83–2.31) | 1.43 (0.79–2.59) |
| **Girls that had never had sex (N = 1,681)** | | | | | |
| Control | 37/843 (4.7%) | 1.00 | 1.00 | 1.00 | 1.00 |
| Intervention | 60/838 (6.6%) | 1.72 (0.88–3.38) | 1.78 (0.90–3.53) | 1.83 (0.92–3.65) | 1.80 (0.95–3.42) |
| **Girls who went to a health facility in the intervention group (N = 1,379)** | | | | | |
| Control | 122/1054 (8.4%) | 1.00 | 1.00 | 1.00 | 1.00 |
| Intervention | 65/325 (17.9%) | **1.90 (1.12–3.21)** | 1.75 (0.98–3.10) | **1.91 (1.08–3.37)** | **2.02 (1.08–3.77)** |
| **Girls who did not got to a health facility in the intervention group^ (N = 1,747)** | | | | | |
| Control | 122/1054 (8.4%) | 1.00 | 1.00 | 1.00 | 1.00 |
| Intervention | 79/693 (10.5%) | 0.91 (0.54–1.55) | 0.89 (0.48–1.62) | 1.17 (0.73–1.90) | 1.30 (0.79–2.14) |
| **Girls who were classified as "near misses"* in the intervention group (N = 1,419)** | | | | | |
| Control | 122/1054 (8.4%) | 1.00 | 1.00 | 1.00 | 1.00 |
| Intervention | 54/365 (15.9%) | 1.23 (0.73–2.07) | 1.20 (0.67–2.16) | 1.45 (0.88–2.40) | 1.52 (0.92–2.51) |

[a] Cluster-level summaries of the geometric means.

All models are adjusted for intention to use contraception at baseline and clustering of girls with robust standard errors.

Model 1 adjusted for age. Model 2 adjusted for age, COVID-19 effect on mobility and currently doing something to avoid a pregnancy. Model 3 adjusted for age, COVID-19 effect on mobility, currently doing something to avoid a pregnancy, city, grade, and wealth quintile. Wald test for all models (unadjusted and adjusted) had a p value <0.05.

*Girls who had an intention to go to health facility (but did not follow through because of COVID, time constraints, lack of support).

^One cluster had no events and was dropped from analysis. Model 1 adjusted for age. Model 2 adjusted for age, COVID-19 effect on mobility, and currently doing something to avoid a pregnancy. Model 3 adjusted for age, COVID-19 effect on mobility, currently doing something to avoid a pregnancy, city, grade, and wealth quintile.

We also examined whether the impact of (re)solve was different among girls who went to a health facility and among girls who reported that they intended to go to a facility but had not yet been able to do so (i.e. "near misses"). When we restricted the sample among girls who went to a health facility in the intervention group, we found that girls in the intervention schools were almost twice as likely to express an intention to use contraception in the next three months compared to all girls in the control schools (aOR = 2.02, 95% CI 1.08–3.77 – **Table 4**). We did not find any significant impact of (re)solve on intention to use contraception among girls in the intervention group who did not go to a health facility and among girls classified as "near misses."

We looked at whether the impact of (re)solve on intention to use contraception differs by responses to attitudes (**Figs B-G in S1 Text**). We only found that intention to use contraception in the next three months differs significantly between girls in the control and intervention schools who said they felt confident to use contraception secretly (or who agreed or disagreed with the statement that they felt confident to use contraception secretly) (p = 0.024).

Finally, given our hypotheses, we looked at whether the impact of the intervention differs by grade and school type (**Figs H and I in S1 Text**). We did not find that intention to use contraception in the next three months differs significantly between girls in the control and

**Table 5. Demographic information of qualitative participants.**

| Participant Category | Baseline | Endline |
|---|---|---|
| Girls (4éme and 3éme) | **Total: 48** | **Total: 41** |
| | *Bobo*: 24 | *Bobo*: 23 |
| | *Ouaga*: 24 | *Ouaga*: 18 |
| Implementing staff (Game facilitators, Pathfinder staff, Health facility staff) | | **Total: 35** |
| | | *Bobo*: 18 |
| | | *Ouaga*: 17 |
| Key informants | | **Total: 14** |
| | | *Bobo*: 6 |
| | | *Ouaga*: 8 |

At endline, in 17 cases where girls could not be reached or declined to participate again, other girls were purposively selected by school staff at the school attended by the original girl.

intervention schools by grade (P = 0.858); however, we found the outcome differs significantly between girls in control and intervention schools depending on school type. The impact of the intervention was greater in the private schools (i.e. girls in intervention schools were more likely to report intention to use contraception in the next three months compared to girls in the control schools) compared to the public schools (i.e. no difference between control and intervention schools) (P< = 0.001).

**Qualitative findings.** We conducted 48 IDIs with girls at baseline and 41 at endline, 35 endline IDIs with implementing staff, and 14 endline KIIs with stakeholders. At baseline, girls were between the ages of 14 and 18, and at endline, they ranged in age from 15 to 25. **Table 5** presents additional breakdown of participants in qualitative research.

## Experience with the (re)solve intervention

Most girls from the qualitative study component reported enjoying playing the game, learning through play, and interacting with facilitators. A 16-year-old girl in 4ème in Ouaga shared, "*What I liked about this game was the way the facilitators were available to us; they were courteous, they listened to us, and they gave us good advice. They showed us what path to take to avoid pregnancy.*" Participants even requested to play the game again. In the IDIs, many facilitators noted high engagement by the girls during game play and noted that they personally enjoyed the opportunity to act as mentors. One facilitator in Bobo said, "*I was able to build good relationships with the girls. I also liked the organization and collaboration with the other [facilitators].*"

## Experience at health facilities

The game sparked curiosity and encouraged girls to seek more information, and in some cases drove them to visit a health facility. Several girls who participated in the endline qualitative interviews had gone to a facility for SRH-related information or services. Quantitative results showed that intervention girls had more trust in health care workers' treatment of adolescent girls, and qualitative results suggested that girls who visited a health facility overwhelmingly had positive experiences. Almost universally, girls who participated in IDIs shared that they were seen by a provider quickly and without harassment or embarrassment. As one 16-year-old girl in Bobo in 3ème told an interviewer, "*The agents [at the health facility] welcomed me as soon as I presented the passport to them, they gave me a place. . . .I was comfortable, because all the questions were confidential. I felt satisfied.*"

Girls' reasons for visiting the health facility were primarily to get information and ask questions–about menstruation and puberty as well as contraceptive methods. Some were seeking care related to painful menstruation, including one 19-year-old girl in Ouaga in 3ème who said, *"I wanted to know [why] when I am in my menstrual cycle it gives me pain and he [the provider] answered all my questions well."* Other girls went to the health facility to learn more about a specific method or discuss options available to them, but few were interested or ready to take up a method. Some girls visited the health facility with specific questions in mind, or to confirm things they had heard–most commonly that implants and other methods cause infertility–while others were generally curious about what the health facility could offer them. According to a 21-year-old girl in Bobo in 3ème, *"As I had doubts about the implants, I went to a health facility and it was explained to me."* Similarly, a 16-year-old girl in Ouaga in 4ème said she went to the facility because *"I wanted to find out if what we have been told at school is the same [as] what I would hear at the health facility,"* and a 19-year-old girl in Ouaga in 3ème said, *"We went to find out more about what the game had already told us."*

## SRH intentions and attitudes

At endline, the majority of girls who participated in IDIs (29 girls out of 41, or about 70 percent), reported that they currently had boyfriends or romantic partners, with relationships ranging from a few weeks to eight years long, but most averaging around a year. However, mirroring quantitative findings, few girls were sexually active. Overall, about 66 percent of all girls we interviewed had never had sex, and of those who had, many reported that they were not currently sexually active.

Compared to baseline, girls at endline reported more positive attitudes toward contraception in general, echoing what we found in the quantitative survey. At baseline, several girls indicated in IDIs that they thought contraception was only for married women, and one 16-year-old girl in 4ème from Bobo said she does not use contraception because, in her words, *"I'm a good girl."* At endline, however, a much more commonly expressed feeling was that contraception was not relevant–because the girls were not sexually active and did not plan to be in the near future–but that contraception was not inherently bad. One 19-year-old girl in 3ème in Ouaga reflected, *"Contraception! It's for all girls. It's a choice. If you want, you can. . . use it, and if you don't want [to], you leave it. Otherwise it's for every girl. . . . It depends on what you want."*

Qualitative results suggest that the solutions challenged girls' misconceptions about contraception and taught them how and where to obtain medically accurate information. At baseline, a number of girls expressed concerns that contraception would be ineffective or would have lasting negative consequences, including trouble conceiving later, but at endline, most girls articulated that modern methods of contraception could successfully and reliably prevent pregnancy and STIs. A 19-year-old girl in Ouaga in 3ème reported, *"I thought contraceptives weren't safe to avoid getting pregnant. But after the game, that changed."* Facilitators remarked that girls were motivated to learn and ask questions about contraception, menstruation, and sexual health, and a health worker in Bobo said, *"I think the girls are starting to understand. They buy into it. Especially those who have enough information about contraception, they do not hesitate to take up a method."*

However, misinformation and fear—especially of a link between contraception, particularly implants, and permanent infertility—were still reported at endline by many girls, in both quantitative and qualitative data. At endline, a 17-year-old girl in 4ème in Ouaga shared, *"I am afraid. . .of using contraception, because [if I use it,] later on I will have no more children."* These fears remain a significant barrier to contraceptive use, and many girls who say they will never use contraception cite fear of negative side effects as the primary driver of this decision.

Girls articulated vague intentions to use contraception at some point in the future, often after marriage, at a certain age, or after completing school, to space births and avoid unwanted pregnancies. A 16-year-old girl in Bobo in 3ème shared, *"[I will use contraception in the future], because at some point I will have sex, and I will have to protect myself to avoid unwanted pregnancies."* Even so, the majority of girls who were not currently using a method of contraception did not report that they thought it likely they would start to use a method in the near future, often because the girls were not currently sexually active and did not expect to become so soon. When asked if she saw herself using a contraceptive method at any time, one 15-year-old girl in 4ème in Ouaga responded, *"No, not really. . . . because I don't intend to be in a relationship."* This offers one explanation for the lack of statistically significant quantitative results, but may mirror the positive trend towards intervention girls forming these intentions in the future.

## Discussion

The (re)solve intervention showed promise for supporting adolescent girls in 4eme and 3eme to use contraception and potentially avoid pregnancies in two large cities in Burkina Faso. This is an important contribution to the field, which has little evidence on successful school-based SRH programming for adolescents in sub-Saharan Africa, especially outside of the HIV context [8, 21–23].

Overall, girls and other research participants reported high levels of satisfaction with the game and positive experiences at the (re)solve health facilities. Girls enjoyed playing the game, asking questions, and learning through gameplay and facilitated dialogue.

We saw an increase in intervention-school girls' intention to use contraception in the near future—our primary outcome for the evaluation—compared to control-school girls, although the relationship was not statistically significant. We noted improvements over time in other key outcomes, such as attitudes about contraception and beliefs about girls who use contraception among intervention-school girls, as compared to control-school girls. In addition, an unexpectedly large percentage of intervention-school girls (365 of 809, or 45.1 percent) intended to go to a facility but had not yet gone because of obligations or restrictions such as school duties, COVID-19 pandemic-related restrictions, or a lack of means, transport or support. Intervention-school girls who went to a (re)solve facility were more than twice as likely to have an intention to use contraception than girls in the control schools. In addition, there were perplexing results related to school type and grade. Given what is known about private schools in urban areas in Burkina Faso, it could be that girls in private schools are older or have more maturity and experience related to sexual and reproductive health; however, the results from all sub-analyses must be interpreted with caution due to cluster size. Understanding the intersectionality of school type, sociodemographic characteristics of students, and patterns of reproductive health behaviors in Burkina Faso, or within the sub-Saharan Africa region, is an important topic of future research.

That the intervention did not encourage more girls to form intentions to use contraception in the next few months is, in some ways, not surprising. On one hand, intention to use contraception is likely to be most relevant to girls who are currently sexually active. Yet despite informal sources and formative research from our team suggesting that many Burkinabe girls are sexually active, our baseline data did not support this. Other evidence from sub-Saharan Africa suggests low rates of reported sexual activity among youth [24]. Our qualitative data showed that many girls are naïve to relationships with boys, or are in relationships but not sexually active, indicating that contraceptive need is not yet relevant in their lives. It is likely that girls are experiencing varying degrees of intimacy rather than existing in a binary state of either

sexually active, or not. In addition, among girls who reported that they were currently sexually active at endline (N = 222), 49 percent reported having sex only on a monthly basis and another 18 percent reported infrequent sexual activity. Given low frequency of sexual activity, contraceptive intention might not be at the forefront of young girls' minds. In addition, given the intervention ended right as lockdowns related to COVID-19 were in place and girls' mobility was restricted, it is also possible that sexual activity—and along with that, intention formation related to contraception—attenuated.

Our sensitivity analyses found some evidence that the intervention had a greater impact on girls who have not had sex, although it should be noted that results in either group were not statistically significant. Several things could explain this. For example, girls who are sexually active might have already considered using contraception and would therefore be more familiar with such methods compared to girls who are not sexually active. This, in turn, would indicate that the intervention might have made a smaller impact on their intention to use contraception in the next three months. Unfortunately, small cell sizes did not allow for sub-analyses among sexually active girls using contraception. In addition, we believe we have measured the most critical variables in the casual pathway. Specifically, we looked at whether the impact of the intervention on intention to use contraception in the next three months differed by different responses to attitudes related to contraception. Our interaction analyses provided limited evidence that this was the case across a range of different attitudes.

We were encouraged by the results suggesting that the intervention shifted attitudes. We noted much lower percentages of intervention-school girls (compared with control-school girls) with beliefs that contraception causes infertility, that providers stigmatize young girls, and that it is normative for unmarried girls not to use contraception. Similarly, there was a higher percentage of girls at endline in intervention schools (compared to control schools) with beliefs that contraception is an option for them, and who feel confident they can both get and use it. This is an important achievement considering that shyness and embarrassment is a barrier for adolescent access to SRH services [25]. Some of these attitudes, such as having high levels of perceived self-efficacy and/or fewer infertility-related fears, have been associated with outcomes such as intention to use and contraceptive adoption [26, 27]. However, fears and misconceptions are still rampant among school-age girls, especially related to perceptions of infertility with modern methods like the IUD and the stigma associated with contraceptive use [12]. This is reflected in large, multi-country studies exploring the reasons for nonuse [28–31]; these myths and misconceptions are indirectly related to contraceptive use [32]. The sticky belief that contraceptives causes infertility is usually grounded in deeper social norms around proving fertility after marriage and the importance of parenthood [33]. It is unsurprising that fear of future infertility would be resilient to a time-bound behavioral intervention such as ours.

The improvements we saw in contraceptive attitudes—coupled with the trends in intention to use a method of contraception in the next three months—suggest the (re)solve solutions may 'prime the pump'. The well-received solutions appear to pique girls' curiosity about SRH, including contraceptives. Behavior change, such as getting girls to use contraception, or even to hold an intention to do so, is not an overnight process. However, even with this light-touch interventions, we can expect modest behavior change: in reviewing exploratory data we found a significant increase of the percentage of intervention-school girls reporting they went to health facility for SRH information or services, from 6.3% at baseline to 32.2% at endline.

Girls' desire to learn about different types of contraception, side effects, menstruation, and other SRH topics suggests that the solutions might have been able to move them further along in their intention formation. It could be that getting unmarried girls to go to the health facility (many for the first time) to gain SRH information and then ensuring a positive interaction

with the health provider is a gateway to future use of SRH services, including contraceptive uptake. Multi-component interventions with activities that address deep-seated norms and beliefs are still needed [34]; however, there is also need for focused, light-touch interventions that are easily scalable in low-resource settings, much like (re)solve was designed to do.

Future research is needed to understand whether playing the game more than once amplifies intention formation and behavior change, or if the game has more influence on sexually naïve girls (although the programmatic implications of this are challenging). There may be opportunities to expand the solution set to older or younger girls, or girls outside the school system; however additional evaluation work is needed. In addition, given the differences by school type, additional research and consideration should be made for how the intervention might be taken up in private versus public schools.

## Considerations for the evaluation

Several factors should be considered in understanding the results of this evaluation. Generalizability of results may be limited to some in-school girls in larger cities in West Africa. First, although we completed a mapping of schools before data collection to inform our sampling frame, it is possible our final sample does not reflect the true distribution of public/private schools in both cities. This is partly due to selecting schools within the catchment areas of our project health facilities. Our effect modification analysis suggests differential impact by school type; therefore, caution should be taken when generalizing to all schoolgirls in public and private schools. Second, the COVID-19 pandemic arrived as implementation of the game in schools was ending; as a result, given restrictions and according to our monitoring data, follow up at the health facility for SRH information or services was disrupted, which likely attenuated the impact of the intervention on some outcomes. Given eleven percent of girls at endline reported they intended to go to the health facility but did not because of COVID-19, we have evidence that fewer girls went for health-facility visits for SRH information and services than might have gone in the absence of a pandemic. It is likely that, as girls were unable to follow up for SRH care at health facilities, those that remained sexually active without using protection became pregnant.

By design, the implementation of (re)solve was staggered, meaning the team completed gameplaying in Bobo before turning to Ouaga. In addition, 314 girls in Bobo from the endline quantitative survey reported they played the game more than once. Playing the game at a later stage or multiple times might have led to differential impact on outcomes, but as this was not something we set out to evaluate formally, it remains unknown.

Even though our previous behavioral diagnosis data indicated that girls would be sexually active, we did not find this to be true in our quantitative data. This is likely due to the fact that participants tend to feel more comfortable sharing sensitive, personal information—especially about their own sexual and reproductive health—in a qualitative interview setting than quantitatively [35]. Given that our outcome of interest is highly related to sexual activity, our subgroup and stratified analyses were underpowered. Caution should be taken in their interpretation. In addition, due to low prevalence of our outcome of interest, the lack of statistical significance in the relationship between the intervention and the outcome might have been due to power; however, we note that the trend was in the expected direction, and other indicators provide evidence of success. Relatedly, as we used the same model for our analyses with attitudinal and behavioral outcomes and stratified analyses, caution should be used in the interpretation when the number of clusters in these analyses is less than 15 per arm (for example, stratified analyses by type of school).

In addition, obtaining accurate data on sensitive topics such as sexual activity and contraceptive use is challenging [36], and it is possible that our data under (or in some cases, over)

reports on sensitive indicators. To try and counter this, the team spent ample time during training on rapport-building tactics, especially for over-the-phone interviews at endline, to ensure privacy and a safe space for sharing personal information. In addition, we did quality checks throughout endline data collection. However, it is not known the extent that others' presence during phone interviews influenced their answers.

Our CRT design and GEE analysis had several strengths. By randomly assigning the intervention to schools from a sampling frame of schools with similar characteristics, we believe our randomization scheme was successful, and we have confidence in our ability to compare our intervention and control groups. We assumed that the correlation matrix was exchangeable, which was in accordance with assumptions made using CRT data where observations within the same cluster might be correlated, but observations on individuals from different clusters are not correlated [18]. We have also accounted for any imbalances at baseline by adjusting for baseline values. We did this for two reasons: first in order to reduce between-cluster variation in our primary outcome at endline and increase the power and precision of the study, and second to take into account regression to the mean. For example, those with low observed values at baseline are expected to show an increase in observed value at endline even in the absence of any true change, while those with higher observed values at baseline are expected to show a decrease [20]. Including a wide variety of stakeholders in qualitative interviews additionally shored up the CRT evidence and bolstered our ability to interpret both expected and unexpected findings. Finally, the COVID-19 pandemic offered an unplanned opportunity to estimate the resilience of our intervention as a stepping stone to accessing SRH resources at the clinical level, and to test phone-based consent and interview processes with a vulnerable population.

## Conclusion

Given the paucity of rigorous data on what works to enhance contraceptive-related outcomes for adolescents in West Africa, our findings are an important contribution to the literature. With substantial documented barriers to girls' access of SRH information and services in sub-Saharan Africa [24], we are encouraged by the shift this intervention appeared to have in boosting girls' confidence, decreasing negative contraceptive attitudes, and getting girls to visit a health facility, or even to intend to visit a health facility.

Our integrated behavioral solutions are a humble but important contribution to Burkina Faso's efforts to address adolescent health and wellbeing, enable adolescents to make healthy choices, potentially reduce the adolescent fertility rate among unmarried adolescents, and reap the demographic dividend by investing in its most valuable asset: youth.

## Supporting information

**S1 Checklist. Inclusivity in global research.**
(DOCX)

**S1 Text. Tables A-C, Figs A-I.**
(DOCX)

## Acknowledgments

We acknowledge all the participants in the (re)solve intervention and evaluation research. We are thankful for the time and energy they gave to us as we learned about innovative programming for adolescent girls in Ouagadougou and Bobo-Dioulasso.

We thank the Pathfinder International implementation team, especially Aicha Tamboura Diawara, Burkina Faso Program Manager. We also thank our colleagues Lydia Saloucou Zoungrana, Bagnomboe Bakiono, and Ganame Afseta. We are thankful to the REM Africa team, and particularly for the project managers, Kalifa Traore and Ibrahim Saley, who managed the 50+ person team of data collectors at baseline, midline, and endline.

We are thankful for the numerous contributions of our colleagues along the way, including from Marta Pirzadeh from Pathfinder International; Jana Smith from ideas42; and Chimaraoke Izugbara, Anam Bhatti, and Elizabeth Anderson from ICRW.

## Author Contributions

**Conceptualization:** Laura Hinson, Mohamad Ibrahim Brooks, Zakari Congo, Reshma Trasi.

**Data curation:** Laura Hinson, Emily Schaub, Aya Mahaman Nourou, Abdou Arthur Abga.

**Formal analysis:** Laura Hinson, Triantafyllos Pliakas, Emily Schaub, Cecelia Angelone, Mohamad Ibrahim Brooks.

**Funding acquisition:** Laura Hinson.

**Investigation:** Laura Hinson.

**Methodology:** Laura Hinson, Triantafyllos Pliakas, Aya Mahaman Nourou, Cecelia Angelone, Mohamad Ibrahim Brooks, Zakari Congo.

**Project administration:** Laura Hinson, Emily Schaub, Aya Mahaman Nourou, Cecelia Angelone, Mohamad Ibrahim Brooks, Abdou Arthur Abga, Zakari Congo, Bruno Ki, Reshma Trasi.

**Resources:** Laura Hinson, Cecelia Angelone, Reshma Trasi.

**Software:** Laura Hinson.

**Supervision:** Laura Hinson, Abdou Arthur Abga, Zakari Congo, Bruno Ki, Reshma Trasi.

**Validation:** Laura Hinson, Zakari Congo, Bruno Ki.

**Visualization:** Laura Hinson.

**Writing – original draft:** Laura Hinson, Emily Schaub, Cecelia Angelone, Mohamad Ibrahim Brooks.

**Writing – review & editing:** Laura Hinson, Triantafyllos Pliakas, Emily Schaub, Aya Mahaman Nourou, Cecelia Angelone, Mohamad Ibrahim Brooks, Abdou Arthur Abga, Zakari Congo, Bruno Ki, Reshma Trasi.

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
