## [Decision Letter · Decision Letter 0]

27 Apr 2022

PGPH-D-22-00235

Does a school-based intervention increase girls’ sexual and reproductive health attitudes and intentions? Results from a mixed-methods cluster-randomized trial in Burkina Faso

Dear Dr. Hinson,

Thank you for submitting your manuscript to PLOS Global Public Health. After careful consideration, we feel that it has merit but does not fully meet PLOS Global Public Health’s publication criteria as it currently stands. Therefore, we invite you to submit a revised version of the manuscript that addresses the points raised during the review process. 

Be mindful that additional comments from a reviewer are in the attachment. 

Please submit your revised manuscript by . If you will need more time than this to complete your revisions, please reply to this message or contact the journal office at globalpubhealth@plos.org. Please include the following items when submitting your revised manuscript:

We look forward to receiving your revised manuscript.

Kind regards,

Adriana A. E. Biney

Academic Editor

Journal Requirements:

1. Please include a complete copy of PLOS’ questionnaire on inclusiveness in global research in your revised manuscript. Our policy for research in this area aims to improve transparency in the reporting of research performed outside of researchers’ own country or community. The policy applies to researchers who have traveled to a different country to conduct research, research with Indigenous populations or their lands, and research on cultural artifacts. The questionnaire can also be requested at the journal’s discretion for any other submissions, even if these conditions are not met.  Please find more information on the policy and a link to download a blank copy of the questionnaire here: https://journals.plos.org/globalpublichealth/s/best-practices-in-research-reporting. Please upload a completed version of your questionnaire as Supporting Information when you resubmit your manuscript

2. We recommend authors use the COREQ checklist, or other relevant checklists listed by the Equator Network, such as SRQR, to ensure complete reporting. For additional information, please check http://journals.plos.org/globalpublichealth/s/submission-guidelines#loc-qualitative-research 

3. Please update your Competing Interests statement. If you have no competing interests to declare, please state: “The authors have declared that no competing interests exist.”

4. In the online submission form, you indicated that “The datasets used and/or analyzed during the current study are available from the corresponding author on reasonable request.”. All PLOS journals now require all data underlying the findings described in their manuscript to be freely available to other researchers, either 1. In a public repository, 2. Within the manuscript itself, or 3. Uploaded as supplementary information.

5. We noticed that you used “data not shown” in the manuscript. We do not allow these references, as the PLOS data access policy requires that all data be either published with the manuscript or made available in a publicly accessible database. Please amend the supplementary material to include the referenced data or remove the references.

6. Please provide separate figure files in .tif or .eps format only and ensure that all files are under our size limit of 10MB.

Reviewers' comments:

Reviewer's Responses to Questions

**Comments to the Author**

1. Does this manuscript meet PLOS Global Public Health’s publication criteria? Is the manuscript technically sound, and do the data support the conclusions? The manuscript must describe methodologically and ethically rigorous research with conclusions that are appropriately drawn based on the data presented.

Reviewer #1: Yes

Reviewer #2: Yes

Reviewer #3: Partly

2. Has the statistical analysis been performed appropriately and rigorously?

Reviewer #1: Yes

Reviewer #2: Yes

Reviewer #3: Yes

3. Have the authors made all data underlying the findings in their manuscript fully available (please refer to the Data Availability Statement at the start of the manuscript PDF file)?

Reviewer #1: Yes

Reviewer #2: Yes

Reviewer #3: No

4. Is the manuscript presented in an intelligible fashion and written in standard English?

Reviewer #1: Yes

Reviewer #2: Yes

Reviewer #3: Yes

5. Review Comments to the Author

Reviewer #1: 1. Summary of the Research and Overall Impression

This is an evaluation of the sexual and Reproductive Health interventions for the girls in grades 9 and 10 in private and public schools in Burkina Faso. It is stated that there was positive reception of the interventions such that at the end of the intervention implementation, girls aged 14-18 years in the intervention group had positive attitudes towards modern contraceptives. The impact of the intervention is more prominent among the girls who have never had sex which makes it easy because by the time they engage into sexual relationships, they will be fully aware of methods of contraceptives and be free to attend the health services (adolescent friendly).

Methodology including sample size is well described. The results section shows that majority of girls received almost all the interventions and were able to contribute in distributing the health passport for other girls to benefit from the intervention. It is evident that most attitudes towards contraceptive use were significantly associated with the intension to use in the next three months. This was almost the same among girls who have visited health facilities and those who did not visit them due to various reasons.

This is good evaluation that is going to generate evidence on the SRH knowledge, attitudes and behaviors especially in the West Africa. I find the evaluation to helpful to Burkina Faso in that the positive reception of the SRH interventions will play a role in reduction of high teenage pregnancy and other SRH issues such as sexual transmitted diseases. This evaluation will also influence adoption of adolescent friendly health services country wide. The girls who were part of the intervention will also spread the work to other girls and eventually majority will be transformed and consequently contributing to the reduction of high teenage pregnancy.

The interventions are well articulated such that it will be easy to replicate or roll out countrywide.

2. Evidence and Examples

a. Major Issues

i. No Major issues

b. Minor Issues

i. Full title: The authors need to be specific on what intervention are they talking about that increases girls’ sexual and reproductive health attitudes and intentions

ii. Line 173: The authors show that participants aged less than 20 they obtained parental consent. One would just wonder if the age of adults in Burkina Faso is 20 years and above

iii. Lines 177-183: endline data was collected in July 2020 due to covid-19 as it was supposed to have happened in March and I am just wondering if covid-19 could have had an impact on the results.

iv. Line 186: The authors purposely selected 9 health facilities, could there be justification of why purposive sampling, why not random sampling or any other sampling, maybe it can come out clearly in that section.

v. Line 294: they are mentioning Figure 1 and line 299 has an instruction to insert Figure 1, could it be the authors forgot to include the figure?

vi. In the results section authors refer to Figures but I cannot see any mentioned Figures in the document. However, in the end there is a link provided for readers to download the figures and tables referred to in the text. I recommend the Figures to be part of the text as it becomes easier for a reader to refer to the figure when reading the text.

Reviewer #2: General comments: This RCT is a well-presented example of rigorous, in-school interventions recommended by international frameworks such as UNESCO’s International Technical Guidance of sexuality education to provide evidence for designing school-based sexual and reproductive health (SRH) interventions. The mixed-methods approach sought to assess the behavior change effectiveness of a board game on adolescents’ intentions to use contraception, intention to engage with health service facilities on contraception, and other related sexual health issues such as puberty and menstruation, and sexual activity.

Please find below the following minor suggestions to increase strength in some areas of the paper:

1. Pg 7, Lines 146-147- The involvement of health care professionals to provide user-friendly services for adolescent participants of the intervention:

Your intervention sought to engage with health workers to assess young people’s visits to health facilities. However, does this approach not introduce any bias in the attitudes of health workers? Indeed, the evidence elsewhere has established the negative impact of health workers on young people’s utilization of sexual health services, but these health workers are aware of the possible intentions of young people to visit health care facilities, and therefore are inclined to exhibit positive attitudes. I suggest an acknowledgement of this bias as a limitation in the ‘consideration for evaluation’ (P 552 ff).

2. Pg 8. Lines 170-172- A point of note (not to be addressed by author): It is commendable that the authors sought ethics approval from an international review board and consulted with a local ethics board in the country (Burkina Faso), something lacking in some studies.

3. Pg 9. Lines 190-192: This comment is on the selection of the majority of student-participants from private schools as compared to public schools. In some educational systems in West Africa, eg. Ghana and Nigeria, most State/Ministry of Education led sexual health programs are implemented in public basic schools, excluding private schools. If this situation pertains in the Burkinabe basic school system, it may affect some assumptions made at the baseline of the participants in the control groups and intervention groups, since there may be some variation in the student engagement in sexual health programs in the private and public schools. Please justify the selection of students in the private schools against this background, and also respond to this limitation (if applicable) in the 'consideration for evaluation'.

4. P. 10. Lines 201-202:What is meant by this statement: ‘…the top girls on the list being invited to participate’.

5. Pp 24 & 25. Lines 490-499. The authors present very interesting insights to suggest that while young people (in SSA and by implication Ghana and Burkina Faso) are known in the literature to be sexually active, there may be variations in sexual activity (ie. degrees of sexual intimacy). I think this is an important finding and should be emphasized in the discussion.

6. Pg 27 553-560. The authors discuss the extent to which the COVID 19 pandemic disrupted continuity in the intervention and may have resulted in the inability of girls to engage with health facilities. This finding presents an opportunity for the authors to extend the discussion to this conclusion-‘With the onset of the pandemic, there was a discontinuity/ a lack of focus in school-based SRH interventions and many young people did not have access to behavior and information change interventions. This may explain the high rates of adolescent pregnancies following the pandemic’.

This analogy is authentic, considering that the authors intend to generalize findings to West Africa, where the rise in adolescent pregnancies is very high, following the pandemic. This point can be drawn on in the discussions section, and you will be credited for providing a novel contribution to ongoing literature on the pandemic and its negative effects on inequities in health delivery.

The authors are commended for the systematic and logical fashion in the discussion of the results.

Reviewer #3: The sample is not clearly described because there is some missing. See suggestions for flowchart.

The real distribution of school children didn't meet the sample. This imply a weight calculation. This is not done.

Have a look on the distribution of school children by age for the difference with the sample. Adjustment is needed.

This implies some reserves for findings

6. PLOS authors have the option to publish the peer review history of their article (what does this mean?). If published, this will include your full peer review and any attached files.

**Do you want your identity to be public for this peer review?** For information about this choice, including consent withdrawal, please see our Privacy Policy.

Reviewer #1: No

Reviewer #2: **Yes: **Benedict Ekow Ocran

Reviewer #3: **Yes: **Idrissa KABORE

---

## [Decision Letter · Decision Letter 1]

6 Oct 2022

PGPH-D-22-00235R1

Does a school-based intervention increase girls’ sexual and reproductive health attitudes and intentions? Results from a mixed-methods cluster-randomized trial in Burkina Faso

Dear Dr. Hinson,

Thank you for your answers to the first round of reviews for your manuscript submitted to PLOS Global Public Health. While two of the reviewers are satisfied with your answers, we wish to share the remaining concerns of Reviewer 3 (see below - and also comments made in the flow chart of the manuscript).

We look forward to receiving your revised manuscript.

Kind regards,

Nguyen Toan Tran

Academic Editor

Journal Requirements:

2. We noticed that you used "data not shown" in the manuscript. We do not allow these references, as the PLOS data access policy requires that all data be either published with the manuscript or made available in a publicly accessible database. Please amend the supplementary material to include the referenced data or remove the references.

Additional Editor Comments (if provided):

Reviewers' comments:

Reviewer's Responses to Questions

**Comments to the Author**

1. If the authors have adequately addressed your comments raised in a previous round of review and you feel that this manuscript is now acceptable for publication, you may indicate that here to bypass the “Comments to the Author” section, enter your conflict of interest statement in the “Confidential to Editor” section, and submit your "Accept" recommendation.

Reviewer #1: All comments have been addressed

Reviewer #2: All comments have been addressed

Reviewer #3: (No Response)

2. Does this manuscript meet PLOS Global Public Health’s publication criteria? Is the manuscript technically sound, and do the data support the conclusions? The manuscript must describe methodologically and ethically rigorous research with conclusions that are appropriately drawn based on the data presented.

Reviewer #1: Yes

Reviewer #2: Yes

Reviewer #3: No

3. Has the statistical analysis been performed appropriately and rigorously?

Reviewer #1: Yes

Reviewer #2: Yes

Reviewer #3: No

4. Have the authors made all data underlying the findings in their manuscript fully available (please refer to the Data Availability Statement at the start of the manuscript PDF file)?

Reviewer #1: Yes

Reviewer #2: Yes

Reviewer #3: Yes

5. Is the manuscript presented in an intelligible fashion and written in standard English?

Reviewer #1: Yes

Reviewer #2: Yes

Reviewer #3: Yes

6. Review Comments to the Author

Reviewer #1: The authors of this manuscript have addressed all my comments and most of them are clear and I fell the paper is flowing now. The manuscript is technically sound and the conclusions made are well represented and supported by the data in the paper. The manuscript meet Plos global public health publication criteria with the layout, presentation of the results and conclusions.

Data is not yet available but Authors have declared that it will be available upon acceptation for publication. The manuscript is written in standard English and concise, straight to the point. The results are well articulated.

With all these, I believe this will be helpful to Burkina Faso and the programme can be expanded to other parts of the country so that other girls can also benefit from the interventions. I therefore, recommend the manuscript for publication.

Reviewer #2: (No Response)

Reviewer #3: The big problem of the manuscript is a statistical bias. As said in the previous revision, the data need to be weighted seriously.

Sampling

Done For autoweighted Weight

Pub Privé Tot Pub Privé Tot Pub Privé

Centre 8 8 3 16 0,423 2,061

Hauts-Bassins 8 8 5 7 0,593 0,923

Total 32 32

Regarding the weight, it is not possible to have the same results in present manuscript. This embarrassed me. Purhaps, it will be better to have advise with an other reviewer expert in quantitative data.

Statistical bias affects the whole result. As the sample was done, without weighting it is not appropriated to conclude for the whole area.

7. PLOS authors have the option to publish the peer review history of their article (what does this mean?). If published, this will include your full peer review and any attached files.

**Do you want your identity to be public for this peer review?** For information about this choice, including consent withdrawal, please see our Privacy Policy.

Reviewer #1: No

Reviewer #2: **Yes: **Benedict Ekow Ocran

Reviewer #3: **Yes: **Idrissa KABORE

---

## [Decision Letter · Decision Letter 2]

18 Jan 2023

PGPH-D-22-00235R2

Does a school-based intervention increase girls’ sexual and reproductive health attitudes and intentions? Results from a mixed-methods cluster-randomized trial in Burkina Faso

Dear Dr. Hinson,

Thank you for submitting your manuscript to PLOS Global Public Health. After careful consideration, we feel that it has merit but does not fully meet PLOS Global Public Health’s publication criteria as it currently stands. Therefore, we invite you to submit a revised version of the manuscript that addresses the points raised during the review process.

Please address the remaining considerations raised by Reviewer 3.

We look forward to receiving your revised manuscript.

Kind regards,

Vanessa Carels

Staff Editor

Journal Requirements:

Additional Editor Comments (if provided):

Reviewers' comments:

Reviewer's Responses to Questions

**Comments to the Author**

1. If the authors have adequately addressed your comments raised in a previous round of review and you feel that this manuscript is now acceptable for publication, you may indicate that here to bypass the “Comments to the Author” section, enter your conflict of interest statement in the “Confidential to Editor” section, and submit your "Accept" recommendation.

Reviewer #2: All comments have been addressed

Reviewer #3: All comments have been addressed

2. Does this manuscript meet PLOS Global Public Health’s publication criteria? Is the manuscript technically sound, and do the data support the conclusions? The manuscript must describe methodologically and ethically rigorous research with conclusions that are appropriately drawn based on the data presented.

Reviewer #2: Yes

Reviewer #3: Yes

3. Has the statistical analysis been performed appropriately and rigorously?

Reviewer #2: Yes

Reviewer #3: No

4. Have the authors made all data underlying the findings in their manuscript fully available (please refer to the Data Availability Statement at the start of the manuscript PDF file)?

Reviewer #2: Yes

Reviewer #3: Yes

5. Is the manuscript presented in an intelligible fashion and written in standard English?

Reviewer #2: Yes

Reviewer #3: Yes

6. Review Comments to the Author

Reviewer #2: The authors have addressed all concerns raised during the first round of reviews. I am happy to recommend that the manuscript is published in its current form.

Reviewer #3: To your response: I'm not convinced with your answer.

It is known that pupil who attend private schools are more some of who failed in public schools. Because of that there are more aged than others. They are aware of sex experience and have some knowledge on reproductive health. They are more well informed on SRH. This implies that girls from private schools have more attention on SRH. Some schools have more aged girls in grades 4e and 3e.

The sample frame is not well described because girls selection in school and grade is not known. Schools size varied, how girls are selected ? This is a lack for the methodology.

7. PLOS authors have the option to publish the peer review history of their article (what does this mean?). If published, this will include your full peer review and any attached files.

**Do you want your identity to be public for this peer review?** For information about this choice, including consent withdrawal, please see our Privacy Policy.

Reviewer #2: **Yes: **Benedict Ekow Ocran

Reviewer #3: **Yes: **Idrissa KABORE

---

## [Decision Letter · Decision Letter 3]

3 Aug 2023

PGPH-D-22-00235R3

Does a school-based intervention increase girls’ sexual and reproductive health attitudes and intentions? Results from a mixed-methods cluster-randomized trial in Burkina Faso

Dear Dr. Hinson,

Thank you for submitting your manuscript to PLOS Global Public Health. After careful consideration, we feel that it has merit but does not fully meet PLOS Global Public Health’s publication criteria as it currently stands. Therefore, we invite you to submit a revised version of the manuscript that addresses the points raised during the review process.

We look forward to receiving your revised manuscript.

Kind regards,

Jianhong Zhou

Staff Editor

Journal Requirements:

Additional Staff Editor Comments: The revision was reviewed by a new reviewer as the previous reviewers and Academic Editor are unavailable. We apologize for the unexpected longer peer review process. We will aim to proceed on the basis of this new review for the next decision. We look forward to receiving your revised manuscript.

Reviewers' comments:

Reviewer's Responses to Questions

**Comments to the Author**

1. If the authors have adequately addressed your comments raised in a previous round of review and you feel that this manuscript is now acceptable for publication, you may indicate that here to bypass the “Comments to the Author” section, enter your conflict of interest statement in the “Confidential to Editor” section, and submit your "Accept" recommendation.

Reviewer #4: (No Response)

2. Does this manuscript meet PLOS Global Public Health’s publication criteria? Is the manuscript technically sound, and do the data support the conclusions? The manuscript must describe methodologically and ethically rigorous research with conclusions that are appropriately drawn based on the data presented.

Reviewer #4: Yes

3. Has the statistical analysis been performed appropriately and rigorously?

Reviewer #4: Yes

4. Have the authors made all data underlying the findings in their manuscript fully available (please refer to the Data Availability Statement at the start of the manuscript PDF file)?

Reviewer #4: Yes

5. Is the manuscript presented in an intelligible fashion and written in standard English?

Reviewer #4: Yes

6. Review Comments to the Author

Reviewer #4: Abstract:

• Line 13: missing word: Compared (to) girls…

Introduction:

• Sentence structure: Girls or women age(d) 15-19 or 15-24.

• Page 7, line 5: four years after having sex for the first time (median age 18.1), and over a year after getting married (median age 20.9).

• Page 7, Line 13: suggested edit: Nearly all studies included in the review were HIV-focused, and there were almost no school-based interventions focusing on contraception and risk perception around pregnancy for girls from Burkina Faso or the West Africa region in general.

• I think that the introduction is a more suitable place to discuss the (re)solve program than the methods section. Please provide some information about the program components as well as its theoretical underpinning. If there are previous studies discussing the design of the program, refer the reader to them in the introduction. There is no need to provide a lot of detail in that case.

• While intention is a good predictor of contraceptive use among girls and women, it would be great if the references were specific to the Sub-Saharan Africa region, as region-specific factors may be at play regarding the translation of intention into action (e.g.: social support, perceived norms). Additionally, none of the citations are specific to adolescent girls, who are the target of the intervention.

Methods:

• Expand the section on the qualitative interviews and explain the kinds of questions each group of participants was asked and where these questions came from, as well as how participants were selected to take part in the interviews.

• Is there a reason why interviewers were mixed-gendered? I would assume that girls may be too shy to discuss questions of sensitive nature with male interviewers. Was there any analysis to determine whether the interviewer’s gender impacted the responses in any way?

Results:

• Page 21: table 3: I feel confident in my ability to use and get a contraceptive method, if I wanted to (not) get pregnant.

• Table 3: the text of table 3 needs to include more information. Are there any statistically significant differences between control and intervention groups on any of the variables?

• I feel that the qualitative section needs a more in-depth analysis. It would be good to highlight the themes that were common/different across different groups of participants (e.g., girls in public versus girls in private schools, sexually active versus non-sexually active girls) and explain whether the findings are consistent with the quantitative results. Second, there was barely any mention of the themes that emerged from the interviews with key informants or with implementation staff. Overall, the value of the qualitative interviews and how they complement and support the quantitative findings needs to be further emphasized.

Discussion:

• “Given what is known about private schools in urban areas in Burkina Faso, it could be that girls in private schools are older or have more maturity and experience related to sexual and reproductive health”. This point needs more elaboration and references if available. Not everyone is familiar with Burkina Faso’s private schools demographics. Why would private school girls be older or more mature than public school girls? Do girls enroll in private schools when they are performing poorly and repeating 4eme and 3eme grades? Are girls in private schools coming from wealthier families? Are their parents more likely to have a higher level of education than those of girls who are enrolled in public schools? What are some possible explanations for this observation?

• “Even though our previous behavioral diagnosis data indicated that girls would be sexually active, we did not find this to be true in our evaluation data.” What may be the reason for that? Are there any substantial differences between the girls from your diagnosis data and those participating in the evaluation?

7. PLOS authors have the option to publish the peer review history of their article (what does this mean?). If published, this will include your full peer review and any attached files.

**Do you want your identity to be public for this peer review?** For information about this choice, including consent withdrawal, please see our Privacy Policy.

Reviewer #4: No

---

## [Decision Letter · Decision Letter 4]

8 Nov 2023

Does a school-based intervention increase girls’ sexual and reproductive health attitudes and intentions? Results from a mixed-methods cluster-randomized trial in Burkina Faso

PGPH-D-22-00235R4

Dear Dr. Hinson,

We are pleased to inform you that your manuscript 'Does a school-based intervention increase girls’ sexual and reproductive health attitudes and intentions? Results from a mixed-methods cluster-randomized trial in Burkina Faso' has been provisionally accepted for publication in PLOS Global Public Health.

Best regards,

Julia Robinson

Executive Editor

Reviewer Comments (if any, and for reference):

Reviewer's Responses to Questions

**Comments to the Author**

1. If the authors have adequately addressed your comments raised in a previous round of review and you feel that this manuscript is now acceptable for publication, you may indicate that here to bypass the “Comments to the Author” section, enter your conflict of interest statement in the “Confidential to Editor” section, and submit your "Accept" recommendation.

Reviewer #4: All comments have been addressed

2. Does this manuscript meet PLOS Global Public Health’s publication criteria? Is the manuscript technically sound, and do the data support the conclusions? The manuscript must describe methodologically and ethically rigorous research with conclusions that are appropriately drawn based on the data presented.

Reviewer #4: Yes

3. Has the statistical analysis been performed appropriately and rigorously?

Reviewer #4: Yes

4. Have the authors made all data underlying the findings in their manuscript fully available (please refer to the Data Availability Statement at the start of the manuscript PDF file)?

Reviewer #4: Yes

5. Is the manuscript presented in an intelligible fashion and written in standard English?

Reviewer #4: Yes

6. Review Comments to the Author

Reviewer #4: (No Response)

7. PLOS authors have the option to publish the peer review history of their article (what does this mean?). If published, this will include your full peer review and any attached files.

**Do you want your identity to be public for this peer review?** For information about this choice, including consent withdrawal, please see our Privacy Policy.

Reviewer #4: No
